# Feasibility and Efficacy of the Newly Developed Robotic Hybrid Assistive Limb Shoulder Exercises in Patients with C5 Palsy during the Acute Postoperative Phase

**DOI:** 10.3390/medicina59081496

**Published:** 2023-08-21

**Authors:** Shigeki Kubota, Hideki Kadone, Yukiyo Shimizu, Hiroki Watanabe, Masao Koda, Yoshiyuki Sankai, Masashi Yamazaki

**Affiliations:** 1Department of Orthopaedic Surgery, Institute of Medicine, University of Tsukuba, Ibaraki 305-8575, Japan; masaokod@gmail.com (M.K.); masashiy@md.tsukuba.ac.jp (M.Y.); 2Center for Innovating Medicine and Engineering (CIME), University of Tsukuba, Ibaraki 305-0821, Japan; kadone@md.tsukuba.ac.jp; 3Department of Rehabilitation Medicine, Institute of Medicine, University of Tsukuba, Ibaraki 305-8575, Japan; shimiyukig@md.tsukuba.ac.jp; 4Department of Neurosurgery, Institute of Medicine, University of Tsukuba, Ibaraki 305-8575, Japan; watanabe.hiroki.gb@u.tsukuba.ac.jp; 5Faculty of Systems and Information Engineering, University of Tsukuba, Ibaraki 305-8573, Japan; sankai@golem.iit.tsukuba.ac.jp

**Keywords:** rehabilitation, shoulder, robotic rehabilitation

## Abstract

*Background and Objectives*: Although postoperative C5 palsy is a frequent complication of cervical spine surgery, no effective therapeutic rehabilitation approach has been established for postoperative C5 palsy. The purpose of this study was to find evidence confirming the effectiveness and feasibility of robotic Hybrid Assistive Limb (HAL) shoulder exercises for C5 palsy. *Materials and Methods*: In this before-after, uncontrolled case series clinical study, we performed a mean of 11.7 shoulder training sessions using a shoulder HAL immediately after the onset of C5 palsy in seven shoulders of six patients who developed postoperative C5 palsy and had difficulty raising their shoulder during the acute postoperative phase of cervical spine surgery. Shoulder HAL training was introduced as early as possible after evaluating the general condition of all inpatients who developed C5 palsy. Patients underwent shoulder abduction training using shoulder HAL on an inpatient and outpatient basis at 2-week or 1-month intervals. Adverse events associated with shoulder HAL training were investigated. The shoulder abduction angle and power without the shoulder HAL were evaluated before shoulder HAL usage, at every subsequent session, and upon completion of all sessions. *Results*: Severe adverse events due to shoulder HAL training were not reported. After completion of all shoulder HAL sessions, all patients showed improved shoulder elevation, while shoulder abduction angle and power improved over time. *Conclusions*: Shoulder elevation training with HAL in patients in the acute stage of postoperative C5 palsy has the potential to demonstrate improvement in shoulder joint function with a low risk of developing severe adverse events.

## 1. Introduction

Postoperative C5 palsy is a frequent complication of cervical spine surgery manifesting as motor and sensory paresis of the muscles controlled by the C5 nerve, which suddenly impairs shoulder joint elevation [1]. Postoperative C5 palsy is defined as deteriorated deltoid muscle strength by one level in manual muscle testing (MMT) regardless of bicep muscle weakness [2]. Postoperative C5 palsy has been reported to have incidence rates of 2.3% [3], 4.6% [4], and 7–12% [5], and is a frequent complication of cervical spine surgery. Conservative therapy is the preferred treatment for C5 palsy that occurs immediately after cervical spine surgery, and spontaneous recovery can be expected. However, impaired shoulder elevation and palsy can persist for several years in some patients [3]. The reported rates of recovery from postoperative C5 palsy were 67% [3], 71% [6], 72.7% [5], and 41% [7]; in many patients, impaired shoulder elevation persists due to an incomplete recovery. Two studies reported that the mean duration from C5 palsy onset to recovery was 4.1 months (3 days–17 months) [3] and 4.5 months (3–36 weeks) [8], while another study reported that ≥6 months is necessary in patients with a shoulder abduction power of grade 2 or lower, multi-segment paresis, paresthesia, and pain at the time of postoperative C5 palsy onset [9]. Impaired arm elevation interferes with daily life; thus, conservative rehabilitation for postoperative C5 palsy is essential for promoting early recovery and preventing deterioration. Rehabilitation approaches for impaired shoulder elevation due to postoperative C5 palsy generally involve physiotherapy and occupational therapy. Active and passive range of motion of shoulder exercises are typically prescribed to prevent shoulder contracture, while strength training is prescribed for paralyzed and residual muscles. However, no effective therapeutic rehabilitation approach has been established for cervical spine postoperative C5 palsy. Impaired arm elevation linked to postoperative C5 palsy, a common complication of cervical spine surgery, can be disabling; therefore, establishing an effective conservative rehabilitation program for the early recovery and prevention of aggravation is urgently required.

The recent availability of robotic rehabilitation using upper extremity robotic device technology has allowed the wide application of highly repetitive and highly intensive training [10,11,12,13,14,15,16,17]. The Hybrid Assistive Limb (HAL) is a wearable, robotic, movement-assistive device that detects muscle action potentials (bioelectric signals) from the user’s skin surface percutaneously and activates the actuator located lateral to the hip and knee joints to assist the user’s voluntary movements [18]. We have been performing robotic rehabilitation using the novel robotic technology with a single-joint HAL in patients with postoperative C5 palsy and associated upper arm disability [19,20,21,22,23]. Shoulder HAL is a wearable motor-assistance robot that detects muscle action potentials (bioelectric signals) on the skin of the lateral arm, and controls or activates actuators located lateral to the shoulder joint (Figure 1). Shoulder HAL has high sensitivity to muscle action potentials generated from surface electrodes attached to the deltoid muscle, i.e., fine bioelectric signals that would not induce a joint movement score of one or two, thus enabling the assistance of voluntary shoulder abduction in real time.

Photographs illustrating the performance of shoulder HAL training. The shoulder HAL system consists of an actuator for the shoulder joint on the lateral side, control device, handy controller, battery, arm attachments, a covered dressing with an arm-forearm splint, surface electrode sensors on the deltoid, and a foundation.

To the best of our knowledge, this study is the first to investigate the feasibility and efficacy of robotic rehabilitation with shoulder HAL in patients with C5 palsy in the acute postoperative phase. This study aimed to find evidence confirming the effectiveness and feasibility of robotic HAL shoulder abduction exercises for postoperative C5 palsy.

## 2. Materials and Methods

### 2.1. Study Design

This one-arm trial investigated the safety and feasibility of shoulder HAL training in patients with postoperative C5 palsy, with shoulder HAL training undertaken during the acute or subacute postoperative phases of the onset of C5 palsy.

### 2.2. Inclusion/Exclusion Criteria

Patients with (1) shoulder elevation difficulty due to postoperative C5 palsy after undergoing cervical spine surgery in our hospital in 2018–2022; (2) who had the ability to understand the study explanation and express consent or refusal; (3) whose body size was a fit with the shoulder HAL; and (4) who had the ability to undergo physical and occupational therapy postoperatively, were included in the study. Contrarily, patients with (1) inadequately controlled cardiovascular and respiratory disorders; (2) intellectual impairments that limited the ability to understand instructions; (3) moderate to severe joint disorders, including contractures at shoulder joints; and (4) moderate to severe involuntary movements, such as ataxia, or impairments of postural reflexes in the trunk or upper limbs, were excluded.

### 2.3. Patients

Seven shoulders (three right; four left) of six patients who developed postoperative C5 palsy after undergoing cervical spine surgery, with difficulty in shoulder abduction, were examined in this study. The patients’ mean age was 65.5 ± 7.5 years. Of the six participants, five were men and one was a woman (Figure 2). Three patients had cervical spondylotic myelopathy, two had ossification of the posterior longitudinal ligament, and one had synovitis, acne, pustulosis, hyperostosis and osteitis syndrome (cervical kyphosis). C5 palsy onset occurred within 3.6 ± 2.1 days (2–7 days) after surgery (Table 1).

### 2.4. Shoulder HAL Training Program

Shoulder HAL is an adaptation of the single-joint HAL (elbow and knee type) and is attached to the shoulder joint. Shoulder HAL consists of the HAL actuator, control device, arm attachment, surface electrode sensors, and foundation; elastic dressing; handy controller; and the arm-forearm splint (Figure 1). The single-joint HAL is attached to the shoulder HAL foundation, and its position is adjusted such that the actuator is located on the lateral shoulder joint. The original arm-forearm splint is attached to the upper arm, the user’s arm is passed through the HAL arm attachment, and an elastic dressing is used to fix the user’s arm and forearm, the HAL attachment, and the arm-forearm splint. The HAL surface electrodes are attached to the deltoid muscle. Wearing the shoulder HAL system requires the assistance of one person and is worn for 3 min.

Shoulder HAL training was introduced as early as possible after evaluating the general condition of all inpatients who developed C5 palsy after undergoing cervical spine surgery. Patients underwent shoulder abduction (at the shoulder scapular position) training using shoulder HAL on an inpatient and outpatient basis at 2-week or 1-month intervals. The 60 min shoulder HAL abduction training sessions consisted of recording vital signs, pre-assessment, HAL training, and post-assessment. The maximum number of shoulder abduction repetitions was performed at each session depending on the patient’s motivation, level of pain, and degree of exhaustion. Two sessions a week was recommended for inpatients, and one session every one or two weeks was recommended for outpatients; however, the frequency was ultimately decided by the patient.

### 2.5. Outcome Measures

Adverse events associated with shoulder HAL training were investigated to determine the feasibility of shoulder abduction training using shoulder HAL. The shoulder abduction angle and power (deltoid) without the shoulder HAL were evaluated before shoulder HAL (at session 1), at every subsequent session, and upon completion of all sessions (after session 10). Shoulder abduction power was evaluated by manual muscle testing (MMT) and measured using hand-held dynamometers (HHDs, u-Tas F-1, ANIMA Corporation, Tokyo, Japan). Evaluation with HHD was performed in accordance with the method described by Andrews et al. with certain modifications [24]. The patients were placed in a supine position, with the shoulder, elbow, and forearm in neutral positions. The dynamometric sensor was placed 5 cm proximal to the lateral epicondyle of the humerus, and the arm was fixed manually by the examiner with one hand. Subsequently, the patients were instructed to perform maximum shoulder abduction, which was measured thrice to calculate the mean maximum shoulder abduction power. This study was approved by the ethics committee of our university (TCRB18-38). The patients were informed about the aim and design of this study and they provided written informed consent for participation and publication.

### 2.6. Statistical Analysis

The *t*-test or Wilcoxon signed-rank test was used to evaluate the differences between pre- and post-HAL training. Statistical analyses were performed using IBM SPSS Statistics 24 software (IBM, Armonk, NY, USA). The alpha level was set at 5%.

## 3. Results

C5 palsy occurred within 3.6 ± 2.0 days postoperatively, while shoulder abduction training with HAL was initiated within 12.4 ± 5.5 days postoperatively in six patients; all six patients underwent 11.7 ± 3.8 sessions (7–18 sessions; Table 1). About 10–450 repetitions of shoulder abduction movements were performed at each session within a mean of 106.6 ± 92.2 postoperative days. As patients had become familiar with shoulder robotics training, the number of shoulder abduction repetitions increased at the later sessions. Inpatients and outpatients underwent 7.3 ± 1.9 and 4.4 ± 3.1 sessions of HAL shoulder training, respectively.

None of the patients experienced severe adverse events nor developed adverse events, such as shoulder pain, severe fatigue, or abrasion in the affected arm during or after shoulder HAL training. The shoulder abduction angle improved significantly from 36.4 ± 8.7° pre-shoulder HAL training to 140.7 ± 25.1 post-shoulder HAL training (*p* = 0.018; Table 2). After shoulder HAL training, five shoulders of four patients achieved a shoulder abduction angle of 120° or more, while two shoulders of two patients achieved 90° or more. Improvement in shoulder abduction angle was observed over time in all patients (Figure 3). The shoulder abduction power (MMT) significantly improved from 1.4 ± 0.5 (MMT grade 1; 4, grade 2; 3) pre-shoulder HAL training, to 3.3 ± 0.5 (MMT grade 3; 5, grade 4; 2) post-shoulder HAL training (*p* = 0.016), and an abduction power of MMT 3 or higher was achieved in all patients (Table 2). The shoulder abduction power was evaluated using HHD and showed improvement over time after each session of training (Figure 4). The seven patients achieved an abduction power grade of MMT 3 (shoulder abduction 90°) in 3.2 ± 2.7 months postoperatively (1–9 months; 1, 1.5, 1, 3.5, 4.5, 9, and 2 months).

The graph showing the chronological improvement in shoulder abduction angle during shoulder HAL training sessions in all patients.

The graph showing the chronological improvement in shoulder abduction power measured using handheld dynamometer during shoulder HAL training sessions in all patients.

## 4. Discussion

Shoulder elevation training with shoulder HAL was performed in seven shoulders of six patients who developed postoperative C5 palsy and impaired shoulder elevation in the acute postoperative phase immediately after C5 palsy onset. After completion of all shoulder HAL sessions, shoulder elevation improved in all patients, and the shoulder abduction angle and power also improved over time; these findings suggest the effectiveness of shoulder elevation training with shoulder HAL as a novel, potentially effective robotic technology tool for postoperative C5 palsy rehabilitation.

Hirabayashi et al. supported the nerve root injury theory as the primary conjectured theory on the causes of C5 palsy [25]; that is, mechanical compression and pull are the primary causes of this condition. This theory supposes that the palsy is linked to microinjury to the C5 nerve root. The factors that can aggravate or prolong C5 palsy remain unknown; however, we speculated that it is determined by the degree of nerve injury caused by compression and pulling forces. The majority of the selected therapies for postoperative C5 palsy are conservative in nature, and spontaneous recovery can be expected. However, shoulder elevation impairment persists in some patients [3]. As inability to raise the arms can adversely affect the performance of daily living activities, a conservative rehabilitation program for postoperative C5 palsy is essential for early recovery (prevention of delayed recovery) and prevention of deterioration.

The shoulder HAL can detect the muscle action potential around the shoulder joint and provide real-time assistance to induce voluntary shoulder elevation in joints with MMT scores of 1–2. All patients in this study showed improved shoulder abduction power and angle; shoulder elevation also improved after completion of all shoulder HAL sessions. However, our findings naturally point to one question: whether the recoveries from C5 palsy were spontaneous or could be attributed to the shoulder HAL training. Since shoulder HAL was introduced early in the acute post-onset phase of C5 palsy, the recovery was considered spontaneous. Imagama et al. analyzed 1858 cervical laminoplasty patients and reported severe postoperative C5 palsy of the deltoid (MMT 2 and lower) in 43 (2.3%) of them. Of these, 29 (67%) patients achieved complete recovery, while 14 (33%) only achieved incomplete recovery with residual palsy with a mean deltoid MMT score of 3.2 (1–4). This figure includes patients who were capable of performing a shoulder abduction of 90° and higher (MMT 3). However, one of the 14 patients did not recover even after 5 years [3]. This finding shows that impaired shoulder elevation persisted in at least one (2.3%) of the 43 patients with C5 palsy (recovery rate: 97.7%). The recovery rate in our study was 100%, since improvement in shoulder function was observed in all six patients; however, this finding is considered inconclusive due to the small sample size. Hence, further studies with a larger sample size are warranted, and a control group is required in addition to comparison with historical controls.

We also investigated the recovery period from C5 palsy onset. Recovery from C5 palsy was markedly poorer in patients with a deltoid MMT score of two or lower at the onset of C5 palsy. Imagama et al. [3] and Nassr et al. [8] reported that the duration of maximal recovery from C5 palsy occurrence was 4.1 months (from 3 days to 17 months) and 20.9 weeks (from 1 to 104 weeks), respectively. All six patients achieved a shoulder abduction power of MMT 3 (shoulder abduction: 90°) in 3.2 months postoperatively (range: 1–9 months; 1, 1.5, 1, 3.5, 4.5, 9, and 2 months). The time to achieving an abduction power of MMT 3 was somewhat shorter in our patients compared with that reported in previous studies; however, hasty conclusions should be avoided as our sample size was extremely small. Meanwhile, the longest time to achieving an abduction power was 9 months. This patient was the oldest (76 years) with severe multi-segment paresis and had limited elbow flexion (MMT [1]) in addition to poor shoulder elevation. Mild multi-segment paresis, which corresponds to MMT scores of 3–4 was observed in many C5 palsy patients; however, severe multi-segment paresis rarely occurred. Multi-segment paresis can delay recovery [9] and was the cause of delayed recovery in this patient. Hashimoto et al. analyzed 199 patients who underwent anterior decompression and fusion and observed 10 (5.9%) patients who developed severe deltoid weakness (MMT 2 and lower) at the time of C5 palsy onset [26]. Five of the 10 patients recovered completely within 3 months, two achieved an MMT score of four within 6 months, and one achieved an MMT score of three within 6 months and a score of four within 15 months. The remaining two patients did not recover. These two patients had a deltoid strength MMT score of two at 6 months from C5 palsy onset. These findings suggest that patients with a deltoid strength score of MMT 2 at 6 months after C5 palsy onset will never achieve a score of MMT 3 or higher. In one patient whose shoulder abduction ability recovered to MMT 3 within 9 months, a shoulder abduction of 50° and an MMT score of two were achieved at 6 months postoperatively. It is, therefore, important to introduce shoulder elevation training with HAL in the acute postoperative phase of C5 onset, as failure to do so could prevent functional recovery in some patients.

Previously, HAL shoulder abduction training was conducted with two other patients with delayed recovery from chronic C5 palsy. The first patient (Case A, man, 62 years old), we reported, achieved complete recovery of the right shoulder dysfunction 7 months after onset of bilateral C5 palsy, but had persistent left shoulder dysfunction (left MMT deltoid score of 2) [27]. Left shoulder abduction with HAL was performed from the 7th month postoperatively once every 2 to 4 weeks, with a total of 23 sessions. HAL training on the left shoulder was performed for 19 months (from 7 to 26 months postoperatively), and the patient’s left shoulder eventually achieved a 95° abduction (MMT deltoid 3) at 13 months postoperatively and 165° shoulder abduction at 16 months. However, C5 palsy improvement was delayed in another patient (Case B, man, 59 years old). Case B underwent posterior decompression and fusion for cervical spondylotic myelopathy. Bilateral C5 palsy and bilateral multi-segment paresis onset were observed several days postoperatively, and bilateral shoulder dysfunction and impaired right elbow flexion persisted as of 9 months postoperatively (MMT right deltoid 2, biceps 2, and left deltoid 2). Ten right shoulder HAL training sessions were performed at 2-to-4-week intervals in the right shoulder at 9 months postoperatively and in the left shoulder at 12 months postoperatively, with a total of 10 sessions. HAL training was performed for 3 months on the right shoulder (from 9 to 12 months postoperatively) and for 5 months in the left (from 12 to 17 months postoperatively). However, abduction only improved from 30° to 55° in the right shoulder (MMT deltoid 2) and from 35° to 55° in the left shoulder (MMT deltoid 2). Shoulder HAL was introduced in the 7th month postoperatively for Case A and in the 9th month postoperatively in Case B; thus, we speculated that the earlier introduction of shoulder HAL training may generate better outcomes. Lim CH et al. reported that the recovery of patients with C5 palsy with shoulder abduction power of MMT 2 or lower, multi-segment paresis, paresthesia, and pain at the onset of C5 palsy takes 6 or more months postoperatively [9]. Case A had single-level C5 paresis, whereas Case B had multi-segment paresis, which increased the recovery rate of Case B. This difference in therapeutic outcomes may be due to the difference in the method of introducing shoulder HAL training; however, it could also be equally attributed to the occurrence of multi-segment paresis. Perhaps the use of an effective technique for introducing shoulder HAL training could have increased the likelihood of Case B achieving improvement similar to Case A.

Shoulder HAL can detect the muscle action potentials from shoulder abduction (MMT score 1 or 2), enabling the assistance of voluntary shoulder joint movement in real time [19,21,22]. The effectiveness of motion training using shoulder HAL for postoperative C5 palsy lies in the biofeedback training to avoid learning trick motions, such as the shrugging motion linked to excessive trapezius contraction due to impaired shoulder abduction. The neuroplasticity of the brain and spine allows the repeated training of errorless biofeedback motions during shoulder HAL training, a potential rehabilitation method that allows earlier recovery and prevents prolongation and aggravation of paresis. Therefore, shoulder HAL training controls erroneous motor learning, including an overactive trapezius, and improves impaired shoulder elevation due to C5 palsy (MMT 2; smooth elevation with shoulder HAL assistance); the repetition of this exercise utilizes the neuroplasticity of the brain and spine in preventing palsy from becoming chronic or severe.

One member of our research group, Lafitte MN, used shoulder HAL for elevation training in patients with shoulder elevation dysfunction and assessed the muscle activity during shoulder HAL training [23]. According to their results, the trapezius activity during shoulder elevation with HAL in patients with difficulty executing this movement was significantly lower than that when this movement was performed without HAL; these findings were consistent among healthy controls as well. Furthermore, Lafitte MN et al. also calculated the coactivation index (CAI) of the deltoid and trapezius muscles during shoulder HAL training [28,29]. CAI is an indicator of deltoid and trapezius muscle coactivation. The CAI of the deltoid and trapezius muscles was significantly lower with HAL compared with that without HAL, suggesting that shoulder abduction movement with HAL lowers the coactivation of the two muscles. These results further suggest that HAL assistance in shoulder elevation reduces the trick motions of the trapezius muscle to enable quality shoulder elevation training while controlling the shrugging motion. In conclusion, we believe that the use of HAL reduced the trick motions of the trapezius muscle during shoulder elevation training, thereby promoting recovery of shoulder joint function as manifested by a wider shoulder abduction angle and a stronger shoulder abduction power. This study aimed to find evidence confirming the effectiveness and feasibility of robotic HAL shoulder abduction exercises for postoperative C5 palsy. Regarding the effectiveness and feasibility of robotic HAL training, there has never been a report that clearly stated the feasibility and effectiveness of shoulder HAL and collected multiple patients. This study is the first to investigate the feasibility and efficacy of shoulder HAL in patients with C5 palsy in the acute postoperative phase.

Many previous studies have analyzed patients with C5 palsy. However, the prognosis of C5 palsy is still poorly understood, limiting our ability to estimate the time required for initial and complete recovery. Therefore, more data on C5 palsy should be collected, and a broader analysis of the course must be performed to determine the most appropriate indications for shoulder HAL. Future studies should also introduce shoulder HAL training in patients with various stages of C5 palsy in order to determine the optimal timing for starting shoulder HAL training. Shoulder HAL has not been tested on a larger proportion of patients with C5 palsy as, inarguably, the majority of patients with this condition recover spontaneously. Conversely, shoulder HAL should be introduced to patients with C5 palsy with little hope for recovery.

This study has some limitations. This was a single-arm study, and did not determine the pure effect of the shoulder HAL training on patients with C5 palsy. In other words, this study was not a controlled trial and could not compare the efficacy of the shoulder HAL training with conventional rehabilitation or spontaneous recovery from C5 palsy. Improved shoulder abduction angle and power results were also observed, however, due to the small sample size of this pilot study (only six patients with a total of seven affected shoulders), it is necessary to increase the sample size by including various stages of C5 palsy, such as acute and chronic phases. To evaluate the true and specific effect of the shoulder HAL training, the study should be repeated, recruiting more patients with postoperative C5 palsy.

## 5. Conclusions

Shoulder elevation training with HAL in patients in the acute stage of postoperative C5 palsy after undergoing cervical spine surgery has the potential to bring about improvement in shoulder joint function, such as shoulder abduction angle and power, with a low risk of developing severe adverse events. The favorable outcomes of HAL shoulder elevation training suggest the effectiveness of this novel robot-assisted rehabilitation tool for this common postoperative complication.

## Figures and Tables

**Figure 1 medicina-59-01496-f001:**
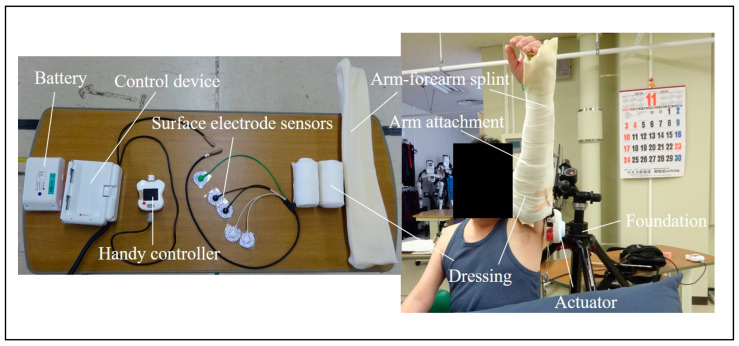
Photographs illustrating the performance of shoulder HAL training.

**Figure 2 medicina-59-01496-f002:**
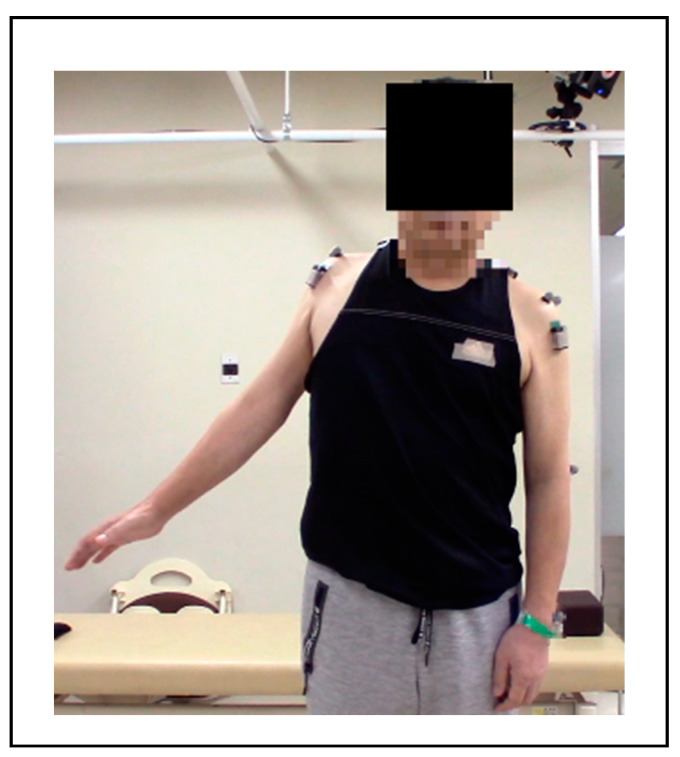
Photograph showing maximum abduction of the right shoulder before shoulder HAL training (Case 2).

**Figure 3 medicina-59-01496-f003:**
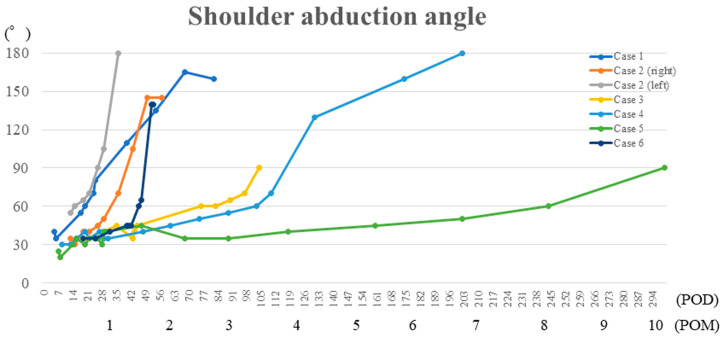
Shoulder abduction angle during shoulder HAL training sessions.

**Figure 4 medicina-59-01496-f004:**
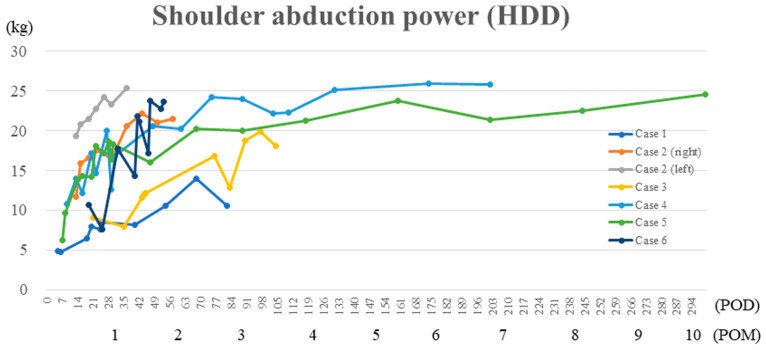
Shoulder abduction power during shoulder HAL training sessions.

**Table 1 medicina-59-01496-t001:** Clinical characteristics of seven shoulders in six patients who received shoulder hybrid assistive limb training.

Case	Age	Sex	Pathology	Operation	Instrumented Fusion Levels	Affected Arm	C5 Palsy Onset	HAL Start After Onset	HAL Finish (POM)	HAL Sessions	Reference
1	65	F	cervical spondylotic myelopathy	PDF	C2-T1	left	4 days	5 days	3 POM	10	[22]
2 (right)	58	M	cervical OPLL	PDF	C2-7	right	2 days	13 days	2 POM	10	[23]
2 (left)			–	–	–	left	2 days	13 days	1 POM	7	[21]
3	59	M	SAPHO syndrome (cervical kyphosis)	ADF, PDF, C6 laminectomy (fibular transfer)	C2-T1	right	2 days	21 days	3 POM	10	NA
4	62	M	cervical spondylotic myelopathy	ADF, ACCF, fibular transfer	C5-6	left	6 days	9 days	6 POM	17	[23]
5	76	M	cervical spondylotic myelopathy	PDF	C2-7	left	2 days	7 days	9 POM	18	[23]
6	73	M	cervical OPLL	PDF, C3-6 laminectomy	C2-7	right	7 days	19 days	2 POM	10	[23]

OPLL, ossification of the posterior longitudinal ligament; SAPHO, synovitis acne pustulosis hyperostosis osteitis; ADF, anterior decompression and fusion; PDF, posterior decompression with instrumented fusion; ACCF, anterior cervical corpectomy and fusion; POM, postoperative months; NA, not applicable.

**Table 2 medicina-59-01496-t002:** Results of shoulder abduction power (deltoid) and shoulder abduction angle before and after shoulder HAL training.

	Before Shoulder HAL	After Shoulder HAL
Shoulder abduction power (deltoid) MMT	1.4 ± 0.5 1; 4 shoulders, 2; 3 shoulders	3.3 ± 0.5 3; 5 shoulders, 4; 2 shoulders
Shoulder abduction angle (°)	36.4 ± 8.7	140.7 ± 35.1

MMT, manual muscle testing.

## Data Availability

The original contributions presented in the study are included in the article. Further inquiries can be directed to the corresponding authors.

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
