# Peer review of "Feasibility and Efficacy of the Newly Developed Robotic Hybrid Assistive Limb Shoulder Exercises in Patients with C5 Palsy during the Acute Postoperative Phase"

_medicina, 2023, doi:10.3390/medicina59081496_

Round 1

Reviewer 1 Report

I have the following comments:

1. The introduction can be improved by the following recent papers:

"Device Design of Ankle Joint Stretching System Controlled by the Healthy Side Ankle Joint Movement for Self-Rehabilitation." Journal of Robotics and Mechatronics 35.3 (2023): 556-564.

"Fixed-time observer-based controller for the human–robot collaboration with interaction force estimation". Int J Robust Nonlinear Control. 2023; 1- 34. doi: 10.1002/rnc.6719.

2. The quality of the figures and table is not good.

3. The results section should be more precise.

4. The conclusion should be improved and detailed.

N/A

Author Response

Reviewer 1

Comments and Suggestions for Authors

I have the following comments:

  1. The introduction can be improved by the following recent papers:

"Device Design of Ankle Joint Stretching System Controlled by the Healthy Side Ankle Joint Movement for Self-Rehabilitation." Journal of Robotics and Mechatronics 35.3 (2023): 556-564.

"Fixed-time observer-based controller for the human–robot collaboration with interaction force estimation". Int J Robust Nonlinear Control. 2023; 1- 34. doi: 10.1002/rnc.6719.

Response:

We added those recent papers that you recommended in the introduction (references 16 and 17).

  1. The quality of the figures and table is not good.

Response:

We improved resolution of all figures and tables.

  1. The results section should be more precise.

Response:

We have been thought that we have shown all the results in the Result section, including C5 palsy occurrences days postoperatively, the starting day of the shoulder abduction training with HAL, the session number of HAL training (Table 1), the number of repetitions of shoulder abduction with HAL per session, the inpatient and outpatient sessions of HAL training, adverse events during HAL training, the improvement of shoulder abduction angle and power (MMT and HHD) before and after HAL training (Figure 3 and 4, Table 2), and HAL training period (postoperatively in months). Therefore, we were able to identify many more results in this study. We sincerely apologize for any oversight or omissions.

  1. The conclusion should be improved and detailed.

Response:

We added slightly detail in conclusion.

“Shoulder elevation training with HAL in patients in the acute stage of postoperative C5 palsy after undergoing cervical spine surgery has the potential to show improvement in shoulder joint function, such as shoulder abduction angle and power, with a low risk of developing severe adverse events. The favorable outcomes of HAL shoulder elevation training suggest the effectiveness of this novel robot-assisted rehabilitation tool for this common postoperative complication.”  (Lines 431-436)

Reviewer 2 Report

This study aimed to investigate the effectiveness and feasibility of using the robotic Hybrid Assistive Limb (HAL) for shoulder exercises in patients with postoperative C5 palsy, a common complication of cervical spine surgery. The study involved 6 patients with a total of 7 affected shoulders. After a mean of 11.7 HAL shoulder training sessions, patients showed improved shoulder elevation, abduction angle, and power without severe adverse events. The findings suggest that early introduction of HAL shoulder training has the potential to improve shoulder function in patients with postoperative C5 palsy.

The paper is well-written and the only concern I have is the small sample size. Although, I do realise the difficulty in obtaining subjects, I would like it to be addressed as a limitation along with the already mentioned various stages of C5 palsy.

Author Response

Reviewer 2

Comments and Suggestions for Authors

This study aimed to investigate the effectiveness and feasibility of using the robotic Hybrid Assistive Limb (HAL) for shoulder exercises in patients with postoperative C5 palsy, a common complication of cervical spine surgery. The study involved 6 patients with a total of 7 affected shoulders. After a mean of 11.7 HAL shoulder training sessions, patients showed improved shoulder elevation, abduction angle, and power without severe adverse events. The findings suggest that early introduction of HAL shoulder training has the potential to improve shoulder function in patients with postoperative C5 palsy.

The paper is well-written and the only concern I have is the small sample size. Although, I do realise the difficulty in obtaining subjects, I would like it to be addressed as a limitation along with the already mentioned various stages of C5 palsy.

Response:

Thank you very much for your valuable feedback. We have incorporated additional sentences discussing the limitations of the small sample size, which were previously mentioned in the discussion section.

"Improved shoulder abduction angle and power results were also observed, however, due to the small sample size of this pilot study (only 6 patients with a total of 7 affected shoulders), it is necessary to increase the sample size by including various stages of C5 palsy, such as acute and chronic phases."  (Lines 423-426)

Reviewer 3 Report

Dear Authors,

Congratulations on the work develoéd. I consider the current version sufficient for publication in this journal.

Author Response

Reviewer 3

Comments and Suggestions for Authors

Dear Authors,

Congratulations on the work develoéd. I consider the current version sufficient for publication in this journal.

Response:

Thank you very much, we appreciate your review.